# High Acid Biochar-Based Solid Acid Catalyst from Corn Stalk for Lignin Hydrothermal Degradation

**DOI:** 10.3390/polym12071623

**Published:** 2020-07-21

**Authors:** Qimeng Jiang, Guihua Yang, Fangong Kong, Pedram Fatehi, Xiaoying Wang

**Affiliations:** 1State Key Laboratory of Bio-based Materials and Green Papermaking Co-founded by Shandong and the Ministry of Science and Technology/Key Laboratory of Pulp and Paper Science & Technology of the Ministry of Education, Qilu University of Technology (Shandong Academy of Sciences), Jinan 250353, China; qimengjiang@foxmail.com (Q.J.); kfgwsj1566@163.com (F.K.); pfatehi@lakeheadu.ca (P.F.); 2State Key Laboratory of Pulp & Paper Engineering, South China University of Technology, Guangzhou 510640, China; 3Department of Chemical Engineering, Lakehead University, Thunder Bay, ON P7B 5E1, Canada

**Keywords:** corn stalk, solid acid catalyst, high acid amount, large specific surface area, lignin degradation

## Abstract

Solid acid catalysts generally show the disadvantage of low acid amount and low recycling rate. To solve these problems, corn stalk-based solid acid catalysts were synthesized through carbonization and sulfonation processes in this work. The results showed that besides the rod-like structure inherited from raw corn stalk, the catalysts contained some small broken pieces on the surface, and the specific surface area varied from 1120 to 1640 m^2^/g. The functional groups (-SO_3_H) were successfully introduced onto the surface of the obtained solid acid catalysts. The acid amount varied between 1.2 and 2.4 mmol/g, which was higher than most of solid acid catalysts. The catalyst produced at 800 °C for 6 h in carbonation and then at 150 °C for 8 h in sulfonation had larger specific surface area and more sulfonate groups. In the degradation of lignin, the use of catalyst led to the generation of more aromatic compounds (65.6 wt. %) compared to that without using the catalyst (40.5 wt. %). In addition, a stable yield of reaction (85%) was obtained after four reuses. Therefore, corn stalk is suitable for high-value utilization to prepare high-acid amount biochar-based catalyst.

## 1. Introduction

Lignin is a renewable aromatic polymer consisting of three phenylpropanoid monomers of p-coumaryl alcohol, guaiacyl, and syringyl units [1,2]. The high value-added products of lignin degradation can be used as fuels or platform chemicals in industry. Different aromatic materials such as vanillin, guaicol, phenols, methoxy phenols, monobenzone and biphenol can be produced from lignin via hydrothermal degradation processes [3,4]. Hydrothermal degradation has been widely studied in biomass conversion with water as the reaction medium. It is efficient in lignin conversion for producing low-molecular-weight compounds, and in the whole hydrolysis process, pyrolysis occurs simultaneously and the degraded products are basically phenolic compounds. It is considered as an environmentally friendly and sustainable technology. Although promising results were reported for production of high value-added products, the production yield and efficiency of the reaction are the main barriers at large scales [5,6]. For now, one option to improve the efficiency of these reactions is to use a solid acid catalyst [7].

Solid acids catalysts are a class of catalysts that loads acids ions onto a solid and provides protons or accepts electrons in a reaction. They can be roughly divided into the following categories: immobilized liquid acids, oxides and metal salts, natural clay minerals, zeolites, heteropolyacids, cationic exchange resins, solid super acids, carbon-based solid acids [8]. Among them, carbon-based solid acids are considered to have potential application value with many advantages of easy preparation, high selectivity, high activity, easy separation and easy recycling [9]. Renewable biomass resources are a better choice for synthesizing carbon-based solid acids via carbonization and sulfonation. In recent years, renewable biomass resources have gained great attention for their ability to synthesize carbon-based solid acid due to their abundance, renewability, biodegradability, biocompatibility and versatility compared to traditional materials for preparing solid acids [10]. Therein, corn stalk is one of the renewable and abundant biomass resources in China, and it contains a considerable amount of carbohydrates, which can potentially be used to produce multifunctional materials. Corn stalk has a strong spongy soft structure, which can be used for immobilizing more functional groups on the surface or inside. Traditional corn stalk-based solid acid catalysts are synthesized from unactivated raw materials, low carbonization temperature or short-time sulfonation, which results in a small specific surface area and a low loaded acid amount [11].

In this study, the solid acid catalysts were produced from corn stalk as shown in Figure 1. The results show that the solid acid catalyst shows rod-like features inherited from raw corn stalk, and it has a relatively rough surface and a distinctly looser structure with more small broken pieces. Notably, the acid amount and specific surface area of the solid acid catalyst are 2.4 mmol/g and 1640 m^2^/g. Furthermore, the hydrothermal reaction can generate more lignin derivatives when the prepared solid acid catalyst is used. The solid acid catalyst also exhibits stable and excellent catalytic activity for lignin hydrothermal degradation even after three runs. Therefore, corn stalk as agricultural waste is suitable for high-value utilization to prepare high-acid amount biochar-based catalyst.

## 2. Materials and Methods

### 2.1. Materials

Corn stalk was collected from Chang Qing district, Jinan (China), washed thoroughly with tap water and air-dried. The air-dried samples were milled using a micro plant milling machine (Tianjin Taisite Instrument Co., Ltd., model F2102, Tianjin, China) at 250 W and 1400 rpm, and sieved to 40–60 mesh. Alkali lignin was also purchased from Tokyo Chemical Industry Co., Ltd. (Tokyo, Japan). Sulfuric acid (95–98 wt.%) and phosphoric acid (85 wt.%) were purchased from Jinan Lan Yu Electronics Co., Ltd. (Jinan, China). and high purity nitrogen gas (>99.999%) was purchased from De Yang Specialty Gases Co., Ltd. (Jinan, China). Chromatographic grade ethyl acetate (99.8 wt.%,) was purchased from Tianjin Damao chemical reagent Co. Ltd. (Tianjin, China). Octafluoro naphthalene (1 pg/μL) was purchased from Agilent Technologies (China) Co., Ltd. (Shanghai, China).

### 2.2. Synthesis of Biochar-Based Solid Acid Catalysts

The preparation process consisted of two subsequent steps: firstly, dried corn stalk sample (10 g) was activated by phosphoric acid solution (100 mL) for 2 h, dried in oven and then carbonized at different temperatures for different time intervals, under N_2_ atmosphere in the tube furnace, based on the conditions listed in Table 1. During the heating process, nitrogen was continuously supplied with a 5 mL/min flow rate in the furnace to ensure an oxygen-free environment for carbonization of the corn stalk. This process was synthesized in a vacuum tube furnace (Zhuochi Instrument Co. Ltd., model SK3-5-12-6, Hangzhou, China); secondly, sulfonate groups were introduced onto the surface and interior of the above carbonized material via sulfuric acid immersion treatment. In this set of experiments, 5 g of the carbonized material produced in the first step was placed in a three-neck flask and 100 mL of concentrated sulfuric acid was added to the flask drop-wise, while the temperature of the system was maintained at 150 °C for 8 h under N_2_ atmosphere. After the reaction, the mixture was cooled to room temperature and then filtered using several layers of filter papers. The product was washed with hot distilled water and then oven-dried at 105 °C overnight for storage reserve. Three different conditions were followed to produce biochar-based solid acid catalysts (SAC) and the final products were labeled SAC-1, SAC-2, SAC-3 as listed in Table 1.

### 2.3. Characterization of Biochar-Based Solid Acid Catalysts

In order to obtain the surface morphology, the sample was coated with a thin layer of gold using an all automatic spray coating machine (SCD 005, Switzerland BAL-TEC Co., Ltd., Neuchatel, Switzerland) and then analyzed by Quanta 200 scanning electron microscope (Holland FET Co., Ltd., Amsterdam, Holland), which operated under an accelerating voltage of 15 kV and vacuum pressure of 200 Pa. This analysis was conducted on raw corn stalk and three prepared samples to identify the impact of carbonation and sulfonation on the apparent morphology.

The existence of different groups on the chemical structure of SAC-1, SAC-2, SAC-3 was validated with a Fourier reflectance infrared (FT-IR) spectrophotometer (Bruker Technology Co., Ltd., mode vertex 70, Karlsruhe, Germany). In this analysis, 0.002 g samples were, respectively, mixed with 0.1 g KBr and then pelletized. The spectra of samples were recorded over a wavenumber range from 400 to 4000 cm^−1^ at 4 cm^−1^ resolution and 32 scans were carried out.

The content of acid sites attached to catalysts was determined by following the titration method. In this set of experiments, 0.05 g solid acid catalyst was added into 10 mL of 0.05 mol/L NaCl solution and the mixed solution was stirred at room temperature for 12 h with a speed of 150 rpm. Then, the suspension was titrated by 0.02 mol/L NaOH solution and phenolphthalein was used as an indicator in this analysis. The content of acid sites was determined based on the consumed dosage of NaOH solution in this analysis [12].

The X-ray diffraction (XRD) measures were performed by a D8 Advance X-ray diffractometer (Bruker Technology Co., Ltd., Karlsruhe, Germany) with Cu Kα radiation (λ = 0.154 nm) at 40 KV and 40 mA to step scan the diffraction angles (2θ) between 5° and 90°. The X-ray photoelectrons spectroscopy (XPS) spectra were obtained by an AXIS Ultra DLD spectrometer (Shimadzu Enterprise ManagementCo., Ltd., Kyoto, Japan) using Mg Kα radiation (hγ = 1253.6 eV) with a step size of 0.1 eV. The thermal stability analysis was obtained by a thermal gravimetric analyzer (TA Q500) with a temperature range from 25 to 700 °C.

In another set of experiments, elemental analysis was conducted using an Elementar Vario EL III (Elementar Trading(shanghai) Co., Ltd., Shanghai, China) for all samples to detect their carbon (C), hydrogen (H), oxygen (O), nitrogen (N) and sulfur (S) contents.

The specific surface area of samples was calculated by a surface area and porosity analyzer (v-sorb2800p, Beijing Jin Aipu Technology Co., Ltd., Beijing, China) using the Brunauer-Emmett-Teller (BET) method within the relative pressure (P/Po) range of 0.05–0.2. The total pore volume was calculated from the adsorption of nitrogen molecules at a relative pressure of 0.99.

### 2.4. Degradation of Lignin Using Biochar-Based Solid Acid Catalysts

Alkali lignin (purchased from Tokyo Chemical Industry Co., Ltd., Tokyo, Japan) was selected as raw material for thermal conversion using biochar-based solid acid catalysts in this set of experiments. At first, 1 g of alkali lignin, 40 mg of catalyst and 80 mL of distilled water were mixed in an autoclave at the temperature of 290 °C and pressure of 8.2 psi for 30 min. After completion of the reaction, the reaction solution containing solid acid catalysts was filtered and washed with deionized water and then dried in oven at 105 °C for 24 h and recycled for reuse. Approximately 80 mL of liquid product of the reaction was dissolved in 20 mL of chromatographic grade ethyl acetate. After filtering, the sample was injected into a gas chromatography-mass spectroscopy (Agilent Technologies (China) Co., Ltd., Shanghai, China, model 5977A/7890B) to analyze its chemical composition. In this analysis, the temperature of the column was 50 °C for 2 min, and then it increased to 160 °C at a rate of 10 °C/min and remained for 2 min at this temperature. Ultimately, the temperature increased to 240 °C at a rate of 20 °C/min and remained for 2 min for completion of analysis. Based on the peak area in the spectra of this analysis, the mass of products generated from lignin degradation was determined. In this analysis, octafluoro naphthalene was used as standard chemicals to calibrate apparatus.

## 3. Results and Discussion

### 3.1. Surface Morphology Analysis

Figure 2 shows the surface morphology of raw corn stalk, SAC-1, SAC-2 and SAC-3. It can be found that the catalyst showed rod-like features inherited from raw corn stalk. These structures remained after the carbonization and sulfonation processes in all samples. In comparison to the raw corn stalk (Figure 2A), the prepared catalysts had a relatively rough surface and a distinctly looser structure with more small broken pieces. It may be attributed to the destructive effect of carbonization and sulfonation; the destroyed components aggregated to form irregular small broken pieces and attached onto the surface of the catalysts, and further formed gaps and holes [13]. More small broken pieces were produced with the increasing carbonization temperature, and it indicates that the structure of the solid acid catalyst was gradually destroyed.

### 3.2. Acid Amount and Specific Surface Area Analysis

The acid amount and specific surface area analysis of solid acid catalysts are presented in Table 2. It is observed that the acid amount in these catalysts varied between 1.2 and 2.4 mmol/g. In addition, the specific surface area of these catalysts varied from 1120 to 1640 m^2^/g. The acid amount and specific surface area were higher than most of solid acid catalysts [13,14,15]. In comparison with this solid acid catalyst, the products in this study had higher sulfonate content and larger specific surface area. Phosphoric acid activation could provide the conditions to form more pores and holes in the carbonization process [16,17]. It is also observed in Table 2 that the catalyst (SAC-3) produced at a higher carbonization temperature and a longer carbonization time had a larger specific surface area and more sulfonate group (compared to other samples). The larger specific surface area of the sample was due to its larger pore volume and pore opening (Table 2). In addition, the destruction of the massive structure and the formation of small broken pieces were also conducive to the increase in specific surface area, as confirmed in SEM images that SAC-3 showed more small broken pieces. Therefore, the solid acid catalyst prepared by a higher carbonization temperature and longer carbonization time could immobilize more sulfonate groups in the next sulfonation process, because the porous structures were in favor of the infiltration and diffusion of acid liquor [18]. These results illustrate that carbonation temperature and time played a crucial role in generating products with a great specific surface area and high acid amount.

The FT-IR spectra of raw corn stalk, carbonized corn stalk and SAC-3 are shown in Figure 3A. Compared to the raw corn stalk, carbonized corn stalk produced at 800 °C contained both aromatic and aliphatic hydrocarbons, which was indicated by the peaks at approximately 2899 (aliphatic C–H stretching) and 894 cm^−1^ (aromatic C–H bending). After sulfonation of the carbonized corn stalk, the most obvious changes are the stretching vibration bands of SAC-3 at around 1210 and 1029 cm^−1^ assigning to the O=S=O stretching of –SO_3_H, which indicates the successful load of the –SO_3_H group on the biochar-based solid acid catalyst [13]. In addition, the stretching vibration band at 1650 cm^−1^ is indicative of the vibration of C=O in the –COOH group [19]. Moreover, the peak at 3428 cm^−1^ was ascribed to the C–OH bending and –OH bending in the –COOH group. This suggests that incomplete carbonization reactions occurred in the carbonization process, resulting in structural changes of the carbon precursor.

In Figure 3B, the XRD pattern of the synthesized solid acid catalysts exhibited a weak but broad diffraction peak at 2θ = 10°–30°, which was assigned to the amorphous carbon and was interpreted as the reflection of aromatic carbon. Moreover, the presence of the plane (100) also indicated a further carbonization during the sulfonation process, which resulted in the growth of the size of aromatic carbon. Different carbonization temperatures even affected the thermal stability of different solid acid catalysts as shown in Figure 3C. From the TG curve, all synthesized solid acid catalysts degraded in two stages. The first stage began at about 25 °C, which lost the weight of about 14.98%, 12.75% and 15.64%, respectively, due to the release of residual and bound water inside the internal structure. The second stage degradation started at about 220 °C. In this stage, the weight loss was about 27.74%, 27.41% and 11.18%, respectively, because of the structural degradation of the solid acid catalysts. In general, SAC-3 showed the lowest degradation rate compared to the other two samples due to the high temperature (800 °C) in the preparing process having carbonized the whole structure.

Figure 3D–F shows the XPS spectra and the surface elemental composition of synthesized solid acid catalysts. As expected in Figure 3D, three solid acid catalysts were dominated by carbon and oxygen peaks, which were the basic elements that made up the constituents of the carbon material. In Figure 3E, the high-resolution spectrum of C1s peak of SAC-3 shows the presence of different chemical states of carbon on the solid acid catalyst surface. The C1s spectrum was fitted into three separate peaks at 284.9, 286.3, 289.0 eV, which are assigned to the C–C/C–H bond, C–O bond, C=O/O–C=O bond, respectively. The O1s spectrum in Figure 3F was fitted into two separate peaks at 533.2 and 531.7 eV, which are associated with C–OH and C=O, respectively. This analysis further confirmed the existence of the –COOH and –OH groups.

The elemental components of raw corn stalk and biochar-based solid acid catalysts are listed in Table 3. It is generally shown that the carbon and hydrogen contents of biochar-based solid acid catalysts dropped compared with the raw corn stalk, while the oxygen and sulfur contents of the samples increased continually from SAC-1 to SAC-3. The drops in carbon and hydrogen are probably attributed to the partial combustion of the material in the carbonation stage, and the increases in the sulfur and oxygen contents are ascribed to the introduction of the sulfonate group on the samples in the sulfonation process. The total concentration of acidic sites determined by acid-based titration was 1.2, 1.8 and 2.4 mmol/g for SAC-1, SAC-2 and SAC-3, respectively (Table 2). In addition, the concentration of –SO_3_H was 0.25, 0.34 and 0.46 mmol/g for SAC-1, SAC-2 and SAC-3, respectively (Table 3).

To evaluate the catalytic performance of the prepared solid acid catalysts, hydrothermal degradation of lignin was performed for producing various degradation compounds. With the above relative analysis, the sample of SAC-3 was proven to be the most suitable choice to verify the catalytic ability of solid acid. The GC-MS chromatogram of liquid product produced from degradation of lignin under two different conditions are shown in Figure 4. The chromatograms of a and b were the products of lignin hydrothermal degradation in the absence and presence of biochar-based solid acid catalyst (SAC-3), respectively. Different compounds would appear in different retention times in this analysis. The phenolic compounds, furfural, ester, acetate, benzene, toluene, heterocyclic compounds and long-chain alkanes were detected at 22.5, 30, 32, 18, 15, 20, 27.5 and 12 min, respectively [20,21,22]. In the presence of the catalyst (Figure 4B), the hydrothermal reaction process was enhanced and the product appeared rapidly after 13 min because of the catalytic effect of the solid acid catalyst. When the degradation of lignin happened, the sulfonic acid group of solid acid catalyst played an important role on activated localization reaction groups and promoted the reaction rate [23,24,25]. Therefore, the products yield and rate were enhanced under the catalytic action of the solid acid catalyst.

Table 4 also lists the mass of products generated from the hydrothermal degradation of lignin with and without the solid acid catalyst. The primary products were aromatic and heterocyclic compounds when the catalyst was used, but without the catalyst under the same conditions, furfural and long-chain alkanes were produced more greatly [21,26]. As a contrast, phenolic and guaiacol compounds from lignin-derived oxygenated compounds were produced by using Ni/ZrO_2_-SiO_2_ catalysts and all of the phenol and guaiacol could be effectively converted into oxygen-free products (cyclohexanol, benzene, methyl-cyclohexane) [27]. This shows the contents of lignin-derived degradation products are various under different catalytic effects of catalysts.

The stability and recyclability of solid acid catalyst are crucial for its practical application. After the first hydrothermal reaction, the solid residue containing catalyst was recovered from the hydrolytic solution by filtration and washing with hot water. After drying, the catalyst was used for the next run at identical conditions. This recycling process proceeded repeatedly for four times. Table 5 lists the properties of SAC-3 after 4 runs of reutilization. Interestingly, the specific surface area of the catalyst hardly changed, but its sulfonate group dropped significantly. This change resulted in a slight reduction in the yield of reaction from 96% after the first run to 85% after the fifth run. These results show that the catalyst can be reused reasonably, but the reuse yield went down gradually because of the recovery loss. The yield stability of solid acid catalyst remained acceptable after three runs; this is because our biochar-based materials have relatively large specific surface area and can continuously immobilize more sulfonic acid groups on the surface and inside. After many runs, there were also enough effective functional groups to continue to participate in catalytic reactions. The results in Table 5 showed that the catalyst was still active in each recycling run, which indicated the stability of the –SO_3_H groups on the catalyst. In addition, the sulfonate group and yield of reaction of SAC-1 and SAC-2 were also determined as shown in Table 6. The sulfonate group of SAC-1 and SAC-2, however, decreased significantly with the increase in the number of recycling times. Therefore, the two samples were not suitable for hydrothermal degradation of lignin. The SAC-3 was chosen to catalyze the hydrothermal degradation of lignin as described in Figure 4 and Table 4.

## 4. Conclusions

In this work, biochar-based solid acid catalysts were successfully synthesized by carbonization and sulfonation from corn stalk. Compared to raw corn stalk, the catalysts had less carbon and hydrogen, but more oxygen and sulfur contents. The hydrothermal reaction could generate more lignin derivatives when the prepared biochar-based solid acid was used as a catalyst. The catalyst also exhibited stable and excellent catalytic activity for lignin hydrothermal degradation even after three runs. The recycling did not affect its specific surface area but decomposed its sulfonate group to some degree.

## Figures and Tables

**Figure 1 polymers-12-01623-f001:**
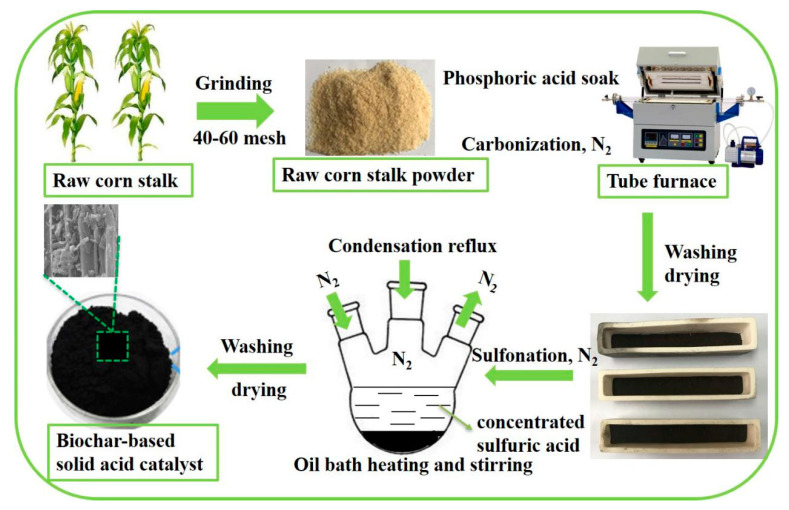
Schematic synthesis of biochar-based solid acid catalyst.

**Figure 2 polymers-12-01623-f002:**
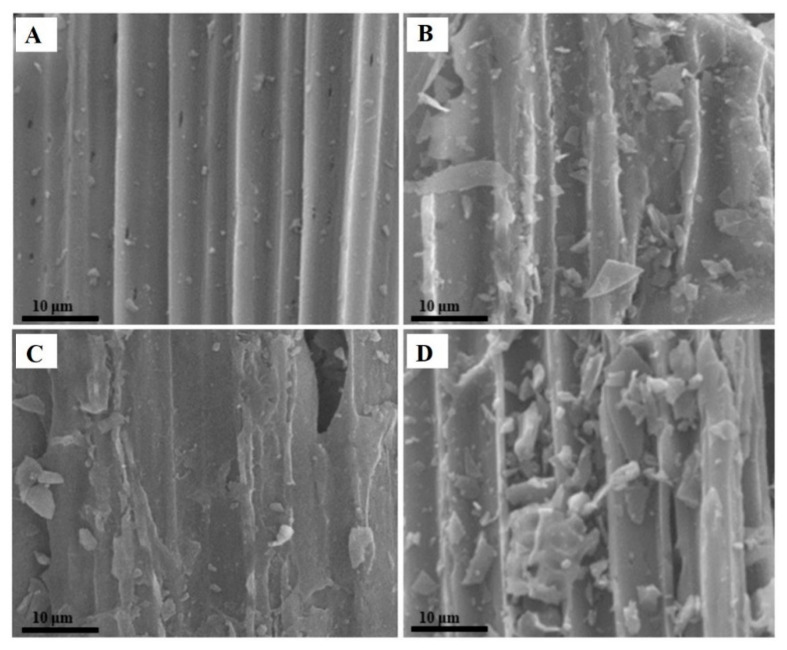
Surface morphology analysis of (**A**) raw corn stalk; (**B**) solid acid catalyst (SAC)-1; (**C**) SAC-2 and (**D**) SAC-3.

**Figure 3 polymers-12-01623-f003:**
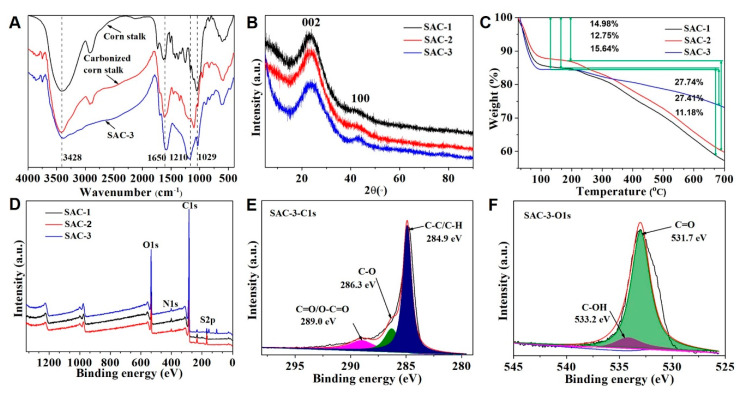
Structural analysis of solid acid catalysts. (**A**) FT-IR spectra; (**B**) XRD analysis; (**C**) TG analysis; (**D**) XPS analysis; XPS spectra of C1s (**E**) and O1s (**F**) of SAC-3.

**Figure 4 polymers-12-01623-f004:**
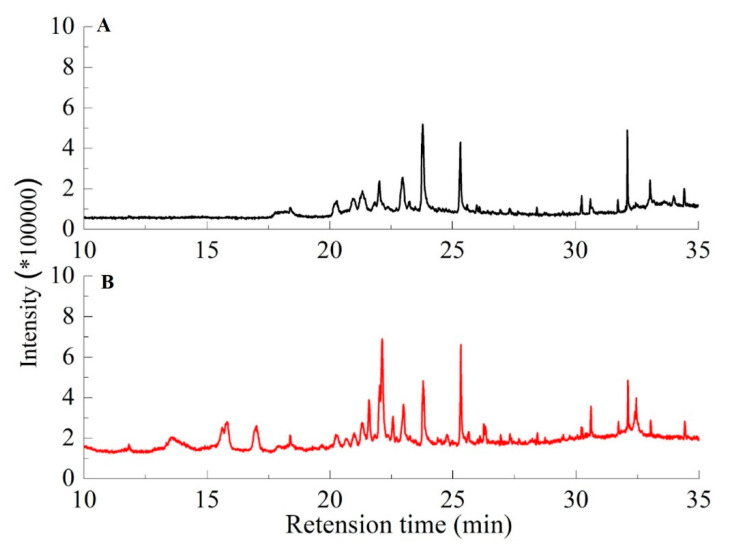
GC-MS chromatogram of products generated from hydrothermal degradation of lignin without (**A**) and with (**B**) solid acid catalyst of SAC-3 under the conditions of 290 °C, 8.2 psi for 30 min.

**Table 1 polymers-12-01623-t001:** Synthesized conditions of biochar-based solid acid catalysts.

Samples	T_1_	t_1_	T_2_	t_2_	Sulfuric Acid Concentration (g/mL)
SAC-1	500	5	150	8	1.84
SAC-2	500	6	150	8	1.84
SAC-3	800	6	150	8	1.84

T_1_: Carbonization temperature, °C; t_1_: Carbonization time, h; T_2_: Sulfonation temperature, °C; t_2_: Sulfonation time, h.

**Table 2 polymers-12-01623-t002:** Acid amount and specific surface area analysis of biochar-based solid acid catalysts.

Samples	BET Specific Surface Area (m^2^/g)	Acid Amount ^a^ (mmol/g)	Pore Volume (cm^3^/g)	Pore Size (nm)
Raw corn stalk	13.9 ± 2	-	0.1	10
SAC-1	1120 ± 5	1.2 ± 0.1	0.685	38
SAC-2	1268 ± 5	1.8 ± 0.1	0.832	46
SAC-3	1640 ± 5	2.4 ± 0.1	1.019	55

^a^ The acid amount of -SO3H groups was determined by titration method.

**Table 3 polymers-12-01623-t003:** Elemental analysis of biochar-based solid acid catalysts.

Samples	C (wt.%)	H (wt.%)	O (wt.%)	N (wt.%)	S (wt.%)	–SO_3_H ^a^ (mmol/g)
Raw corn stalk	72.27	3.41	23.59	0.22	0.16	-
SAC-1	70.92	2.45	24.82	0.24	0.96	0.25 ± 0.05
SAC-2	66.18	2.10	28.26	0.22	1.24	0.34 ± 0.05
SAC-3	64.73	1.86	31.52	0.23	1.64	0.46 ± 0.05

^a^ The density of –SO_3_H groups was determined by elemental analysis.

**Table 4 polymers-12-01623-t004:** Products of hydrothermal reaction of lignin degradation with and without solid acid catalyst.

Products with/without Solid Acid Catalyst	Aromatic Compound (wt. %)	Furfural (wt. %)	Ester (wt. %)	Acetate (wt. %)	Benzene (wt. %)	Toluene (wt. %)	Heterocyclic Compounds (wt. %)	Long-Chain Alkanes (wt. %)
with	65.6 ± 0.5	6.2 ± 0.5	2.4 ± 0.2	1.6 ± 0.2	2.0 ± 0.2	1.8 ± 0.2	14.2 ± 0.5	6.2 ± 0.2
without	40.5 ± 0.5	34.7 ± 0.5	0.6 ± 0.2	1.4 ± 0.2	0.4 ± 0.2	0.5 ± 0.2	11.7 ± 0.5	10.2 ± 0.2

**Table 5 polymers-12-01623-t005:** Effects of number of reuse times of SAC-3 on specific surface area, sulfonate group and yield of reaction.

Number of Recycling	BET Surface Area (m^2^/g)	Sulfonate Group (mmol/g)	Yield of Reaction (%)
0	1640 ± 5	2.4 ± 0.1	--
1	1605 ± 10	2.0 ± 0.1	96 ± 1
2	1565 ± 12	1.7 ± 0.1	92 ± 1
3	1514 ± 15	1.2 ± 0.1	89 ± 1
4	1410 ± 15	0.4 ± 0.1	85 ± 1

**Table 6 polymers-12-01623-t006:** Effects of number of reuse times of SAC-1 and SAC-2 on sulfonate group and yield of reaction.

Number of Recycling	SAC-1		SAC-2	
	Sulfonate Group (mmol/g)	Yield of Reaction (%)	Sulfonate Group (mmol/g)	Yield of Reaction (%)
0	1.2 ± 0.1	-	1.8 ± 0.1	-
1	1.0 ± 0.1	95 ± 1	1.5 ± 0.1	95 ± 1
2	0.7 ± 0.1	90 ± 1	1.1 ± 0.1	91 ± 1
3	0.4 ± 0.1	86 ± 1	0.6 ± 0.1	87 ± 1
4	0.3 ± 0.1	82 ± 1	0.4 ± 0.1	84 ± 1

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
