# Peer review of "High Acid Biochar-Based Solid Acid Catalyst from Corn Stalk for Lignin Hydrothermal Degradation"

_polymers, 2020, doi:10.3390/polym12071623_

Round 1

Reviewer 1 Report

This manuscript provided useful solid acid catalyst with enhanced lignin hydrothermal degradation. The prepared solid acids from corn stalk were characterized by SEM, BET, FTIR, XRD, TGA, XPS and EA analysis. They concluded that a higher carbonization temperature and time could immobilized more acid group due to porous structures. To evaluate the catalytic activity of the prepared solid acid, the hydrothermal degradation of lignin was performed and detected by GC-MS. The results indicated that different degrade products were found in the presence and absence of solid acid. The recycling used test was also studied, showing reasonable recycling used ability. Overall, this study was interesting and could be accepted after minor correction.

  1. How to define yield of reaction in Table 5?
  2. Please show the yield of reaction in the absence and presence of raw corn stalk, SAC-1 and SAC-2.

Author Response

Thnaks for your review. We have provided a point-by-point response to the reviewer’s comments. Please see the attachment.

Reviewer 2 Report

I reviewed  article " High acid biochar-based solid acid catalyst from corn stalk for lignin hydrothermal degradation " by Jiang et al.

This is a well designed and presented experimental article. Lignin is an important polymer resource to be utilized for different applications and this work tries to illustrate  that.  Presentation of the article is good. I have one minor comment,  it would be good idea to have a clear statement as justification at the end of Introduction .

Figure 3 has six small subfigures  and some of the fonts are just too small perhaps they can be revised so that the readers can have a better visualization. Otherwise article is ready to be published.

Author Response

Thanks for your review. We have provided a point-by-point response to the reviewer’s comments. Please see the attachment.
